# A Review of the Potential Health Benefits of Low Alcohol and Alcohol-Free Beer: Effects of Ingredients and Craft Brewing Processes on Potentially Bioactive Metabolites

**Duane D. Mellor \*, Bishoy Hanna-Khalil and Raymond Carson**

Aston Medical School, Aston University, Birmingham B15 2TT, UK; b.hanna-khalil@aston.ac.uk (B.H.-K.); r.carson@aston.a.c.uk (R.C.)

**\*** Correspondence: d.mellor@aston.ac.uk

**Abstract:** Beer is a beverage of significant historical and cultural importance. Interest in the potential health effects of alcoholic beverages has largely focused on wine; however, there are a number of potentially beneficial bioactives that beer may contain that warrant further investigation. The challenge of considering any potential health benefits of beer are restricted by the negative consequences of its alcohol and energy content. There is potential to enhance the bioactive qualities of beer whilst reducing the alcohol and energy content through novel brewing approaches often used in craft brewing, in terms of ingredients, brewing methods and type of fermentation. Consumer demand to produce a greater variety of beer types, including alcohol-free beers, may also help to increase the number of beers which may have greater potential to improve health, with lower levels of alcohol, while still being tasty products. As low alcohol, prebiotic and bioactive containing beers are developed, it is important that their potential health benefits and risks are fully assessed.

**Keywords:** beer; alcohol; polyphenols; phytoestrogens

## 1. Introduction

The role of beer in society is one of history, debate and even myth, with some believing beer, as it is boiled during production, was used as a safe alternative to water in medieval times. Although it has become part of folklore, it is perhaps not grounded in reality, as the availability of water from safe sources was not as rare as many believe, and water supplies were often transported through towns and cities by relatively advanced systems [1]. Although sewerage may have been less well developed, references to public toilets in cities such as London date back to the thirteenth century. It is apparent that safe water was available; however, it appears to be the case that it was not the primary beverage of choice.

Beer and, in particular, what were known as "small beers", which were the product of the third brewing or running of the mash (with the first running producing the strong ale, the second, common ale and the final, the small beer) was commonly consumed by both adults and children from at least the eleventh century in the UK [2]. The history of brewing beer as part of human activity dates back to ancient Egypt and Mesopotamia. Therefore, if it was not consumed for hygiene reasons, why was it consumed? The reason perhaps was that it was an effective method of maintaining hydration. The combination of carbohydrate and electrolytes may facilitate the absorption of water, as well as providing valuable additional dietary energy (calories), which was incredibly important at a time when energy expenditure was higher, with the typical laborer expending 12.6 Mj (3000 kcal per day),

significantly more than their modern counterpart. Historically, it has been suggested that the alcohol content of beer inhibited the growth of bacteria and pathogens, providing a safe alternative to what many consider were unclean water supplies. This belief is likely to not be correct, as the yeasts used in brewing at the time were far less effective than those typically used in modern brewing, resulting in beers with very low alcohol contents. Therefore, any benefits may have been due to their electrolyte and carbohydrate contents, making them similar to modern sports drinks, something that the marketing of a number of modern alcohol-free beers are starting to exploit [3]. It is plausible that these traditional weak beers contained other metabolites that could have contributed to beneficial effects, but this would purely be speculation.

For centuries, beer production was a small-scale industry, although ales were transformed in the Middle Ages by the addition of ingredients such as hops, which added new flavors, alongside potentially bioactive compounds that may have helped as preservatives and antimicrobials. It was not until the industrialization of the industry that things changed significantly. In some parts of Europe, for example, Germany, brewing became regulated with legislation via the Purity Laws (Reinheitsgebot) introduced in 1516 [4], which restricted the ingredients that could be used in the production of beer. This reductionist approach to brewing could have had the potential to reduce the variety of potentially bioactive compounds in beer. Although it also enhanced quality and at a time when the adulteration of food and beverages was not uncommon, it also possibly made beer safer for the public. The one key change which likely reduced heterogeneity in beer during industrialization was the movement to a reduced number of yeasts and the emergence of *Saccharomyces cerevisiae* and related strains, e.g., *S. pastorianus* that, from the nineteenth century, became the dominant organisms used in brewing globally [5].

In the last couple of decades, there has been a move against homogenous beer production with a growth in demand from consumers for variety in beer, with craft and micro-brewing becoming increasingly popular. In the USA in 2018, craft beer sales grew at a rate of 4% by volume, whereas there was a decline of 1% overall for beer volume sales [6]. In the UK, the Society of Independent Brewers Association reported an increase in craft beer production of 0.8% amongst its members [7]. The effect of this has been to increase the range of ingredients and a renewed interest in different yeasts and fermenting bacteria to deliver new flavors for an increasingly demanding consumer [8]. This review aims to consider the potential health effects of these novel brewing processes, both with respect to bioactive compounds derived from the ingredients, produced through the brewing process, and the probiotic effect of the yeasts and other microorganisms responsible for fermentation.

Along with growing interest in the potential benefits of compounds and organisms found in beer, it is important to consider the potentially negative aspects of beer consumption, which are predominantly associated with the alcohol and energy content. The risks of alcohol consumption are well established, with the recognition moving from the view that moderate (<14 units/week) alcohol consumption may be associated with reduced risks of early mortality and, especially, cardiovascular disease to a more contemporary view that there may be no safe threshold for alcohol consumption [9,10]. Therefore, when considering the potential health benefits of beer, careful consideration should be given to novel brewing techniques and yeasts that could help to moderate or even minimize alcohol content whilst optimizing the flavor and potential health effects. With respect to the energy content, this again will be considered, as will how the formulation and the effect of fermenting microbes can influence the amount of carbohydrate, especially residual sugar, that will remain in the final product. Other considerations are the influence of the recipe formulation, brewing conditions and fermenting microbes on the formation of compounds known as "congeners", which have been linked to the symptoms of headaches and hangovers. Therefore, it is essential to consider the balance of the potentially beneficial effects of beer with the known negative consequences associated with the habitual consumption of alcohol at even relatively low levels [10,11].

The challenge is to show how the bioactives (naturally occurring from hops and fermentation, secondary metabolites from yeast and lactic acid bacteria) can be maintained in a product that is acceptable to consumers but without the negative aspects of additional dietary energy (calories), sugar and alcohol. In producing such products, it needs to be clear that there is a market for them

which appears to be the case given the increase in alcohol-free beers, some of which are being marketed as isotonic drinks with health benefits due to their vitamin content, including $B_{12}$, alongside their polyphenol content [3]. Technical issues include how best to produce these products using different strains of microorganisms, which may create various challenges including varying lengths of fermenting time; higher residual carbohydrate levels, as this is not used as a substrate for ethanol synthesis; or the consequences of brewing to a lower alcohol by volume (ABV), which may inhibit the development of flavor. An alternative approach would be the removal of ethanol after fermentation. This approach can tend to have undesirable effects as it can result in cooked flavors if heat is used to evaporate the ethanol. The effect may go beyond taste, as heat may, in addition to removing alcohol, be likely to denature potential prebiotics and polyphenols.

## 2. Materials and Methods

A narrative review of the literature was undertaken including literature from laboratory and human studies alongside data from food science literature to develop an overall critical picture of the potential benefits and risks of beer consumption and, especially, craft and micro-brewery produced beers with respect to the bioactives contained in the ingredients or produced through the brewing process, along with a review of the effects of using different fermentative microbes. This was triangulated with public health recommendations and evidence, so that the benefits were balanced, in a pragmatic and responsible way, with the risks of alcohol and excess dietary energy intake.

The analysis of the literature will take an iterative approach to develop plausible theories and hypotheses to suggest how brewing might be developed to produce products that may be able to maximize any theoretical benefits and minimize potential adverse effects.

## 3. Bioactives from Ingredients and Brewing Processes

Beer at its most simple and in accordance with German Purity Laws can be limited to only four ingredients: barley, hops, water and yeast. The addition of yeast to this list came years later, once its role in the fermentation process was understood. Globally, different grains or carbohydrate sources are used, from wheat, rye, oats and sorghum, all of which can be malted, to rice and maize, which can be added to the mash in a flaked form. Much of the attention on bioactives contained within the ingredients has focused on the hops (*Humulus lupulus*); however, a number of grains may also contain compounds which could have potentially beneficial effects on human health including vitamins, minerals and phenolic compounds such as phytoestrogens. One example is that barley grown in some regions is a source of selenium, which is a useful antioxidant in the body and may be more important than was previously thought [12].

### 3.1. Hops

Hops have been used in brewing since the ninth century, but were not grown and therefore not widely used in brewing until much later in the UK, during the fifteenth century. Hops were used initially for their preservative properties; however, over time, the bitterness and aroma imparted to the beer became appreciated, but additionally, hops have been used as part of traditional medicine. The potentially beneficial effects of hops have been linked largely to their polyphenol content, with in vitro data [13] suggesting effects as antioxidants and in reducing adipocyte differentiation and enhancing nitric oxide production. The levels of these compounds will be highly variable, due to the differences between hops, the times of harvest, and further alterations induced by processing and brewing.

The potential for hops as a source of bioactives, including polyphenols, is considerable [14], leading to a number of potential targets that may be affected via hormonal mechanisms, as some of the bioactives can mimic estrogen, whilst others may function via antioxidant mechanisms or through altering metabolism to enhance the metabolism of xenobiotics and even regulate lipid metabolism. This has led to considerable interest in the hop, not only for its flavor and aromatic properties in beer, but as a potential nutritional supplement. This is because hops, which generally refers to the female

flower cones of the plant, have been found to be a rich source of prenylated phenols, including chalcones and flavonoids (some of which are classed phytoestrogens) [14]. It has been proposed that the estrogen-modulating effects of hops and their extracts are due to the effects of them being antagonists and binding to the estrogen receptor, preventing the action of estrogen, as well as the potential to inhibit aromatase, which is the rate limiting step in the synthesis of estrone and estradiol ($E_2$). This has led to the emergence of hop-based products as potential supplements to support women going through the menopause [14].

Additionally, it has been proposed that hops contain potential antimicrobial compounds that have been associated with a group of 56 phenolic compounds [15], with a range of potential activity, including inhibiting virus replication and bacterium and fungus growth as effective antimicrobials. This could help with the shelf-life of the beer as a product, as well as having the potential to influence health either within the gastrointestinal tract or systemically, depending on bioavailability. The potential health benefits of hops are likely to be due to the complex and unique nature of the organic compounds they contain. These compounds have been reported to demonstrate atypical oxidation behaviors that could have implications both with regards to how they are used in beer production and in relation to health benefits. However, the compounds which may invoke health effects are not necessarily the same as those which impart flavor, which is their traditional purpose [16].

In vitro analysis of one class of polyphenol found in hops, phytoestrogens, has shown them to influence the way breast cancer cells in the laboratory aggregate, grow and invade [17,18]. This appears to be via the effect of hop phytoestrogens on an E-cadherin dependent mechanism. Whilst these types of observation are scientifically interesting, as they highlight a clear potential mechanism of how hop phytoestrogens can influence the physiology of cancer cell lines, the clinical applicability of this is questionable, as the influence of alcohol on breast cancer risk and the effect of the dietary energy on the risk of obesity, which is an additional risk factor for breast cancer, puts into doubt any potential benefit of hops consumed in beer for breast cancer risk or management. Whilst the hypothesis of pathway modulation by polyphenols is the most plausible, owing to the low doses and poor bioavailability of most polyphenols, the older antioxidant theory, based largely on in vitro work, still dominates much of the literature [19]. The theory has merits if the enhanced antioxidant capacity relates to up-regulation of liver pathways such as increasing glutathione levels; however, the potential for mass free radical or reactive oxygen species scavenging is largely dismissed as a plausible mechanism [20]. In the case of polyphenols in beer, this is somewhat more challenging, as the polyphenols may be metabolized during the brewing process, which may enhance or reduce any potential activity. Additionally, if the mechanism is via either interaction between the polyphenols and the gut microbiota or the stimulation of liver enzyme pathways, this could potentially be inhibited by the hop polyphenols being co-ingested with alcohol.

Hops are also a source of trace elements such as zinc and manganese, which are required by a number of enzyme pathways in the body, including those having a role in protection from reactive oxygen species, such as via mitochondrial manganese superoxide dismutase [11]. Recently, there has been a focus on the manganese content of hops. Although the manganese content is lower than the iron content, owing to how these ions are leeched into the beer, the transfer of manganese from the hops is significantly greater [21]. This has implications for how the micronutrient profile of beer can be maximized, as manganese—like other divalent metal ions, e.g., copper and iron—can negatively impact flavor stability. This suggests that there needs to be a compromise with respect to how and when hops are added to beer to maximize any potential nutritional benefits without adversely impacting on flavor or shelf-life [22].

### 3.2. Beer as a Source of Prebiotics and Probiotics

Beer, as one of the oldest fermented foods, has been largely overlooked in the recent increase in interest in this area, with respect to both gut and general health [23,24]. Although there are some epidemiological studies and reviews of the effects of the consumption of beer on mortality, most of the focus has been on wine [25]. The reasons for these benefits with respect to reduced mortality, largely linked to reduced risks of cardiovascular disease, have mainly focused on the potential of

moderate alcohol consumption or bioactive compounds from the hops or other ingredients, e.g., sprouted cereals such as malted barley. This largely ignores the potential of the microbes used that are responsible for the fermentation and how these may influence human health. This potential could manifest itself in a number of ways:

- Providing a source of probiotic microbes, which requires a viable dose of live organisms to be in the product when it is consumed. This would not be the case in pasteurized canned or bottled beers, nor in filtered products; filtration may significantly reduce microbial loads depending on the nature of the filtration process.
- Providing prebiotic metabolites, through the secondary metabolism of compounds derived from the grain, hops or other ingredients. The effects of fermentation have been well described in the production of sourdough breads but less so in that of beer, although the fermentation processes are intrinsically similar, with the exception that these types of bread tend to involve both yeast and lactic acid bacteria; a practice that is only typical in beers such as the lambic and American wild ales.
- The production of antimicrobial compounds, including ethanol, alongside a range of compounds, depending on the recipe and type of microbes involved in the fermentation process [26].

The effect of industrialization in the nineteenth century and the demand for more homogenous products led to the development of brewer's yeast, which consists of a limited range of species of *Saccharomyces*, with *S. cerevisae* being typically used in ales and *S. pastorianus* (previously known as *S. carlsbergensis)* used in lager [5]. These yeasts have become dominant, especially *S. cerevisae*, due to their alcohol tolerance and rapid rates of fermentation, which are ideal for commercial production. However, the increasing interest in craft brewing and consumer demand for product diversity has led to the exploration of various microbes, both yeasts and bacteria, in the brewing process. This, akin to many of the developments in modern beer production, is largely driven by commercial reasons and the desire to develop new flavors; however, its potential to improve health could be seen to be reflected in how fermented products have been used historically and continue to be used by a number of indigenous populations globally. In parts of Africa, a number of staple grains are spontaneously fermented by yeasts and bacteria to produce a range of products. These products range from those produced from millet, sorghum and maize into beverages that would be considered to be beer, through to products specifically used to manage diarrheal diseases [27,28]. The role of such beers has been considered to be both economic, providing incomes to small producers and others involved in its production and distribution, and nutritional, as it may increase the supply of energy and other nutrients, typically B vitamins, in areas prone to under-nutrition. There has even been a suggestion that it may reduce the risk of exposure to aflatoxins [29].

### 3.3. Analysis of Polyphenols and Phenolics in Beer

A number of studies have investigated the characteristics of beers with respect to the phenolics found in the final product. The majority of the antioxidant capacity (55%–88%) in commercial beers appears to be derived from six compounds, with ferulic acid representing over 50% of this and the others being syringic acid, (+)-catechin, caffeic acid, protocatechuic acid and (-)- epicatechin [30]. However, this work was limited to commercial lagers, which lack the complexity of many craft beers and ales. The number of phenolics found in beers varies according to both the recipe (the amount of barley malt and hops) and, perhaps more interestingly, the place origin of the beer. However, this may be expected as the growing conditions of plants are known to influence the production of these (potentially protective) secondary metabolites [31], or to be byproducts from the different brewing processes used [16]. A study of various lagers and ales that were mostly non-hopped beverages found that Czech lagers had more phenolics than other beers and that these were mostly derived from barley (malt) [32], with 47 different compounds being identified. This diversity was not seen in Indian lagers, brewed using yeasts that sink during fermentation, potentially reducing the potential for secondary metabolite formation [33]. Where beers are made from other ingredients that are

significant sources of polyphenols, such as fruits, these unsurprisingly have higher levels of phenolics. In a study which analyzed both fruit and non-fruit beers, cherry beers, followed by fruit beers, had the highest levels of polyphenols, with catechins being the most dominant compound detected [34]. The variety of the polyphenols in fruit beers could be, in part, due to the additional sugar contained in the fruits, which facilitates secondary fermentation, especially in lambic and sour beers. However, these additional polyphenols appeared to be associated with a higher alcohol content and potentially sugar content, although that was not reported in this study. The higher alcohol content is likely to negate any potential benefits that the polyphenols may provide. The bioavailability of polyphenols has been considered a potential barrier to any benefits [20]. Although dark beers vary in their phenolic properties, in general, the color tends to be associated with the total phenolic concentration [35]; it appears that a number of beer phenolics, especially cinnamic acid, may be hydrolyzed under alkaline conditions, which potentially mimic the environment of the small intestine. This implies that more work is required to consider not just the concentration and range of polyphenols in beers, but also how the polyphenols found in beers are metabolized and absorbed in the human digestive tract. It has been estimated that around 57 phenolic and 11 nitrogenous compounds have been identified in craft beer, with 12 of these being only identified in beers [36]. There is some capacity to use some of these molecules to differentiate between craft beer styles, such as caffeoyl- and coumarolyquinic acids, coumaric acid, kaempferol-3-O-rutinoside and proanthocyanidin B dimers III and V, which can differentiate IPA, lager and weiss beers [36].

Early work on the phenolic characteristics of low calorie and non-alcoholic beers suggested that these had lower levels of polyphenols [37]; however, this study was carried out before the growth of hopped low alcohol beers, which are brewed and produced in different ways. Further analysis of emerging products is required, both to assess potential health benefits and to substantiate any attempted health claims. The potential of more contemporary non-alcoholic beers, with respect to both the phenolic and bitter acid components (alpha-acids or humulones and beta-acids or lupones), has been explored in mouse models of Alzheimer's [38]. It is possible that the bitter acids have been studied less due to their labile nature and the poor solubility of lupones. Despite these limitations, mechanistic effects of these compounds have been shown via PPAR gamma, which have the potential to moderate glucose and lipid metabolism as well as to reduce cognitive decline.

### 3.4. Influence of Fermenting Organisms

The potential of some yeasts to act as probiotics is an area of developing interest; although not uncommon amongst brewers, the likely effects of these are only starting to be explored. *Saccharomyces boularidii* is a potential candidate as a probiotic yeast [39], which tends to produce less alcohol, but similar sensory qualities, with the commercially desirable characteristics of high viability at lower pH, which helps to maintain the shelf life of the end product beer. *S. boulardii* is very similar to *S. cerevisae* and can only be distinguished by advanced typing methods that investigate differences in metabolism and physiology [40]. There is emerging evidence of the potential of *S. boularidii* [41] in both the prevention and management of gastrointestinal diseases, which is hypothesized to be due to its ability to colonize, produce antimicrobial compounds and support healthy mucosal and immune system function. These include the in vitro potential of *S. boularidii* via the production of small peptide metabolites, that have inhibitory effects on a number of pathogens including *C. difficile* and *V. cholera*, alongside other metabolites that have been identified as having potentially tropic and immunoregulatory effects towards the host [40]. This mechanistic potential has been linked to potential clinical applications including antibiotic related diarrhea and acute gastroenteritis in children [42]. In relation to the benefits of *S. boularidii* [43] in brewing, these include its ability to survive in the final beer for up to 45 days and higher antioxidant capacity, with similar polyphenol content to commercial beer but a much lower alcohol content, suggesting that different yeast strains could both enhance potential health benefits and minimize the potentially negative aspects associated with beer, namely alcohol content.

Although modern brewing has tended to use a limited number of yeasts commercially, this was not the case historically, and some specialist beers, particularly lambics, use spontaneous

fermentation, which depends on naturally occurring microbes. This is a lengthy process and initially yields high levels of enterobacteria before yeasts and lactobacilli are established and alcohol and lactic acid concentrations start to rise, which can take 1–3 months. This is before other bacteria start to colonize, which then lead to the finishing of the beer and its flavor profile. Industrially produced sour beers tend to avoid the enterobacteria by adding lactic acid to the wort, which is cooled before being added to wooden barrels that then inoculate the beer with bacteria [44]. This type of brewing tends to result in a microbiome of several species of yeast and bacterium, rather than the near monoculture of a typical industrially produced lager or ale. *S. cerevisiae* and *S. pastorianus* are commonly found in lambic beers, as are *Dekkera bruxellensis* (and its anamorph *Brettanomyces bruxellensis*) and *Pedioccus damnosus*. Although *D. bruxellensis* is associated with spoilage in wine, it is associated with the finishing flavor of sour beers and often exists symbiotically with the lactic acid producing *P.damnosus*. The potential of such mixed microbiomes in fermented products has been hypothesized to be part of their benefits, and possibly why single strains of probiotics may be less effective [45]. *Brettanomyces* species are well known for being able to produce a very low final gravity in beers by breaking down the residual dextrins, which *S. cerevisiae* is not able to do. Therefore, a beer fermented with added *Brettanomyces* would have a low residual carbohydrate content.

It is therefore important to consider the fermenting organisms which are used to inoculate the wort, as these can have a significant impact on the potential health benefits of the end product, both in terms of additional function with respect to health and the moderation of the alcohol content, which is considered to be the key negative component of beer when it comes to health. The effects of different organisms on prebiotic production are less well explored in a brewing context. They have been assessed theoretically for some yeasts, e.g., *S. boularidii*, in terms of enhancing adaptive immunity and regulating inflammatory signaling effects [41]. There is also evidence with *S. boularidii* of trophic activity on the intestinal mucosa and "cross-talk" with host microbiota [41]. This is unlikely to be due to direct signaling, but via the production of secondary metabolites, which is the principle of dietary prebiotics. The concept of microbes producing metabolites that have health benefits via such an indirect pathway is a novel area but is logical and warrants further investigation.

During mashing in the brewing process, starch in the grains is converted to maltose and dextrins, with the yield of each varying according to the mash temperature. Dextrins are oligosaccharides that are known to act as prebiotics in the colon [46]. There is increasing interest in the role of prebiotics in health, and the potential of beer as a source of prebiotics is underrated and under-developed. These can additionally reduce the dietary energy and sugar contents of the beer. It has been speculated that these oligosaccharides, that are not available as substrates to the human digestive tract, are in some cases produced as secondary metabolites by fermenting organisms and may have prebiotic functions, providing substrates for commensal organisms in the colonic microbiota. Whether these oligosaccharides exist in the ingredients of the beer or are produced during the process of brewing is an area of potential development with a number of non-conventional yeasts such as *Saccharomyces cerevisiae* var. *diastaticus* and *Dekkera* spp [5].

A significant challenge with respect to considering craft beer as a source of live bacteria and yeasts, and therefore as an effective probiotic, is balancing the beneficial aspects of flavor maturation through secondary fermentation as part of conditioning in the bottle with spoilage. However, some studies report viable counts of microbes in unfiltered beer products being adequate to meet the requirements to be permissible for probiotic claims in Japan. The nature of these microbial ecosystems may vary and is likely to change through the brewing and maturation process, as especially the aerobic species are likely to die soon after bottling [47]. This perhaps will mean that either probiotic beers will have a relatively short shelf life or that they will require refrigerated storage, perhaps using fermenting organisms to produce secondary metabolites that have either direct or indirect prebiotic activities; this could be a more practicable solution.

### 3.5. Effects of Brewing and Maturation Processes

Hops are known to contain polyphenols; however, the method used in brewing will influence the amount and types of polyphenols found in the end beer. A number of polyphenols are associated

with bitter flavors, and often, brewers debate the optimal time and method of adding hops to the beer. This can range from adding them into the mash prior to boiling, through to dry hopped beers where the hops are added later during fermentation, e.g., in the last 2–7 days, through to methods that infuse beers with hop extracts at the point of consumption via a Randall. These approaches will all have the potential to vary the amount of any potential beneficial compounds in the end beer.

The effect of brewing itself has two possible ways in which it can influence the levels of potentially beneficial compounds in the end beer product. Firstly, the production of alcohol and increasing alcohol levels can help to dissolve more compounds from the hops, although a higher alcohol content may not be desirable for health. The other potential effect is seen with prolonged fermentation, where secondary metabolites may be produced. The value of the hops being added during the brewing process is that they are antibacterial and so can reduce spoilage. This may enhance the growth of organisms that may benefit health, either indirectly via metabolite synthesis (prebiotic) or directly via colonizing the gastrointestinal tract (via a probiotic mechanism). The levels of polyphenols and their antioxidant capacities in vitro are apparently stable throughout the brewing process, with higher levels being apparent in the end beer [48], which is explained by the theories stated above.

## 4. Intervention Studies

A number of small studies have sought to investigate the potential health benefits of beer, including alcohol-free beers. A small randomized clinical trial of 33 males aged 55–75 years of age who drank moderately, with a high risk of cardiovascular disease, were given either beer, non-alcoholic beer or the equivalent amount of gin in an open study. Giving beer or alcohol-free beer with 1208 mg polyphenols reduced overall cardiovascular risk markers; the addition of 30 g of alcohol in the beer increased HDL cholesterol and adiponectin whilst reducing fibrinogen. The alcohol-free drink additionally reduced systolic, but not diastolic blood pressure compared to the other beverages [49]. Alcohol-free beer, in a single arm study of 29 nuns aged 53–73 years for 45 days, did not decrease markers of inflammation but reduced oxidized LDL cholesterol and increased glutathione levels, suggesting an enhanced antioxidant capacity to potentially reduce the risk of cardiovascular risk [50]. Additional work investigating the effect of dealcoholized beer on hemostasis, which was part funded by the brewer, in 12 young men, showed it to reduce both PAI activity and Factor VII coagulant activity, which could potentially reduce the risk of atherosclerotic disease, at least in the short term [51]. Other intervention studies have sought to alter the carbohydrate source in the beer, using resistant maltdextrin and isomaltulose in an alcohol-free beer in 41 individuals with prediabetes or type 2 diabetes. The altered carbohydrate source resulted in improvements in glycaemia; however, there were some gastrointestinal side effects related to the osmotic effects of this non-digested carbohydrate [52]. There appears to be emerging evidence from clinical trials that alcohol-free beers may have modest short-term effects on markers associated with cardiovascular disease; however, the potential negative effects of residual sugar in the products needs to be considered, as this has been associated with adverse health consequences [53]. As of February 2020, a search of the World Health Organisation (WHO) Clinical Trial Registry (https://www.who.int/en/) identified eight registered clinical trials, of which only three have been published, two have been included in this review [49–52] and the third is related to bowel function, which was not within the scope of this review.

## 5. Challenge of Energy and Sugar Content of Beers

The alcohol content of beer is a justifiably significant concern if potential health benefits of beer are being considered. Alcohol is the leading cause of liver disease [54] in more economically developed countries and is associated with significant levels of social dysfunction, reduced productivity and poor health [55]. Therefore, before considering a credible case for beer having health benefits, the level of alcohol in products needs to be reduced, which is not without challenges, as low alcohol beers have been associated with inferior flavor and aroma profiles [56]. Strategies to meet the demand from government public health policies and consumers have led to greater interest in alternative brewing methods and alternative yeasts, as physical processing methods to reduce the

alcohol content of beer tend to reduce the volatile flavor compounds. A number of candidate yeasts including *Saccharomycodes ludwidii*, *Zygosaccharomyces rouxii* [57] and *Saccharomyces boularidii* [58] have been assessed as candidates, producing a beer that is acceptable to consumers, commercially viable and low in alcohol. There are additional interests in yeasts such as *S. boularidii*, as this also has probiotic potential. Additionally, the potential of some yeasts to reduce sugar content, including *Saccharomyces cerevisiae* var. *diastaticus* and *Dekkera*/*Brettanomyces* spp. due to their ability to digest dextrins, could further reduce the sugar and energy contents of beers [5]. However, this needs to be balanced with the effects of these yeasts on flavors, with a tendency of the *Dekkera*/*Brettanomyces* spp. to produce esters that can result in off flavors.

Currently, under European Union regulations, alcoholic beverages with more than 1.2% absolute volume of alcohol (ABV) are exempt from the obligation to list ingredients and nutrition information (EU No. 1169/2011) [59]. However, in March 2015, the Brewers of Europe committed to bringing in the labelling of ingredients, including allergens, and to start to list nutritional information by 2022 [60]. Prior to this, it was only necessary to state the volume, ABV and, additionally, the number of units of alcohol. Typically, beers do not contain any additional added sugar; however, the use of malting results in an increased amount of energy and carbohydrate, especially in the form refined carbohydrates and sugar linked to the effects of yeast, and can, in some beers, result in a significant sugar content. The energy content of beer is primarily due to the alcohol content (29 kJ per gram; 7 calories per gram) and carbohydrate. The amount of carbohydrate is variable, depending on the type of beer. Only a few types have added sugar (e.g., some fruit beers, lactose and sour beers), but it is mostly derived from the malted cereals added to the wort at the start of brewing and the action of yeast, which can vary from very low levels of sugar in pils lagers (although these can be higher in alcohol) through to porters and stouts, which can have upwards of 20 g of carbohydrate per pint. This, akin to the alcohol content, can be an issue of matching the flavor, aroma and, potentially, the color profile of the product. Therefore, in addition to the alcohol and energy, sugar content needs to be considered if a beer is to be considered a health product, with a number of the non-conventional yeasts having the potential to reduce the sugar content of novel beers.

Although, until recently, beer production has tended to focus on producing a "clean" rapid ferment that produces alcohol in the most commercially viable way, this view is being challenged from a public health and consumer perspective. Where public health policy is advocating for reductions in energy and alcohol contents, consumers are often demanding greater variety and flavor. These two goals may not be as divergent as they may first appear, in that by using different strains of yeast and bacteria, it may be possible to reduce alcohol and energy contents whilst enhancing any potential health qualities.

The data on the nutritional composition of beer are largely based on data that are greater than 10 years old [37,61,62]. This is likely to mean that contemporary craft beers and, especially, modern alcohol-free ales that can be considerably "hopped" (e.g., Brewdog Nanny State and Hambleton Ale Point 5) would not be included, and more up to date analysis is required. Table 1 highlights that beer has the potential to provide significant amounts of a number of nutrients, including being a potentially vegan source of vitamin $B_{12}$ along with folate and—depending on the type of malt, hops and brewing and finishing process—a range of polyphenols. This needs to be balanced against the alcohol, energy and sugar contents. The processes used for alcohol-free beer production eliminates the alcohol, and in addition some of these products may be low in energy and sugar too, however this is not always the case. When compared to a range of common beverages, it may have the potential to be a better source of many vitamins and minerals than milk or orange juice.

**Table 1.** The nutritional composition of beer and a comparison with other common beverages. The data are from [61–63].

| Nutrient | Range in Beer (Per 100 mL) | Amount in Common Beverages Per 100 g |
|---|---|---|
| Energy (Kcal) | 15–110 | Milk 35–70 |
| Carbohydrate (g) | 0–6.1 | Milk 5.0 |
| Sugar (g) | 0–6.1 | Orange Juice 8.5 |
| Protein (g) | 0.3–0.5 | Milk 3.5 |
| Vitamin C (mg) | Up to 30 | Orange Juice 22–48 |
| Riboflavin (mg) | 0.002–0.08 | Milk 0.23 |
| Niacin (mg) | 0.3–0.8 | Milk 0.2 |
| Vitamin $B_6$ (mg) | 0.007–0.17 | Milk 0.06 |
| Folate (mcg) | 4–60 | Milk 8 |
| Vitamin B12 (mcg) | 0.3–3 | Milk 0.9 |
| Sodium (mg) | 4–23 | Orange Juice 3 |
| Potassium (mg) | 33–110 | Orange Juice 164 |
| Iron (mg) | 0.01–0.05 | Milk 0.2 |
| Zinc (mg) | 0.001–0.148 | Milk 0.5 |
| Selenium (mcg) | Up to 0.72 | Milk 1.0 |
| Polyphenols(mg) | 12–52 | Black Tea 104.48 |

## 6. Discussion

Although beer is a food product with one of the longest histories, it has received less attention with respect to its potentially positive effects if consumed in moderation when compared to wine. This may be in part related to the association of wine with affluence and status, whereas beer has been linked with overconsumption and the working class. The composition of beer has a number of potential benefits, including its B vitamin content from the yeast, which is one of the few non-animal sources of vitamin $B_{12}$ (which is not within the scope of this review [64]). However, its alcohol content and relatively high energy content linked with heavy consumption is associated with a severe health cost both to individuals and society as a whole.

With industrialization, there was a move towards the production of large amounts of uniform products, including for beer, which led to a reduction in the variety of beer processes and ingredients, including yeasts, used in its production. Recently, there has been a growth in demand for craft beers, causing a significant growth in different brewing methods, cultures used to ferment the beer and ingredients. Subsequently, there is a re-emergence in the potential of beer to be a food stuff that could contribute to improved health. However, it is unlikely that different beers will have the same potentials to positively influence health. A beer that is hopped or slowly fermented with a probiotic yeast with a low alcohol content may be very different in terms of its health benefits to a lager with a high alcohol content.

Studies have shown that the moderate consumption of alcohol is associated with a lower risk of mortality compared to abstinence or excessive consumption. In the case of beer, it is possibly the effect of consuming beer, similar to consuming other fermented plant-based foods [64], in terms of its polyphenolic composition and how that interacts with the host's gastrointestinal microbiome. The benefits of beer consumption at moderate levels have been reported as being similar to wine in reducing cardiovascular disease risk [65,66]. Although the risk of alcohol is well described, the effects of reducing the alcohol and, potentially, dietary energy in beer whilst retaining the polyphenols, vitamins and minerals in a fermented beer product is unknown. It has been suggested that beer containing significant amounts of folate and choline may have benefits on the human gastrointestinal microbiome, which in traditional beers is potentially compromised by the alcohol, which is toxic to many gastrointestinal organisms. There is emerging evidence that the consumption of beer as well as wine may have beneficial effects on the gastrointestinal microbiome, enhancing levels of *Lactobacillus* sp. and *Bifidobacterium* sp [67]. Having stated the lack of experimental data, it is a logical deduction

based on its composition that it could have the potential to meet the requirements for health claims, if it is alcohol-free.

It seems apparent that, if not for the alcohol, many beers could be considered to be effective health drinks, with the potential to be at least as effective as a number of commercially available sports drinks as rehydration solutions. However, there are a number of claims regarding some beers that have become part of folklore, including the iron content of stout. Prior to concerns of the association of alcohol with congenital abnormalities and fetal alcohol syndrome, it had been recommended to pregnant and nursing mothers. The benefit was more likely to be due to its high content of energy and not its iron content, which is only around 2% of a woman's daily requirement per pint. However, as many dark beers are brewed using hard water, it can contain up to 8% of an adult's daily requirements, per pint, of calcium, which, alongside the electrolytes and sugars, can help with the absorption of the water in the beer. Historically, beer was made in copper vessels and, indeed, the vessel used to boil the wort is still called a "copper". Most modern breweries use stainless steel equipment now; however, there are still large numbers of copper breweries in Germany and the Czech Republic. Recent thinking has suggested that copper could be involved in some important reactions during brewing and that some of the copper will leach into the beer. Some brewers with stainless steel equipment are now adding copper metal during brewing. Copper is an important trace element required by certain enzymes in the body, and a source of copper may be beneficial for health [22, 68]. A case-control study on beer drinkers in the Czech Republic found the lowest risk of myocardial infarction in men who drank beer daily (4–9 L per week); however, the apparent protective mechanism is not clear. Moderate alcohol consumption has been linked to lower rates of cardiovascular disease; however, it has been suggested that the copper content of beer may be responsible for this beneficial effect [68].

The botanical compounds, which are largely derived from the hops, are perhaps the most studied potentially beneficial elements found in beers. These vary between different types of and brewing methods used for beer. Additionally, polyphenolic compounds can be derived from the grains, which are typically sprouted during malting, which can increase the levels of these antioxidant compounds. It is not entirely clear if these compounds act directly to influence health or if they are metabolized by the fermenting organisms in the brewing process or by the gastrointestinal microbiome, or even a combination of these processes. Although the exact mechanisms of how any potential health benefits of beer consumption occur are unclear, it appears that the moderate consumption of beer may have health benefits, and craft beers containing higher levels of these compounds, probiotic yeasts and bacteria and lower levels of alcohol may have additional benefits.

Perhaps one of the biggest challenges to the potential of beer products being considered healthy is how they are consumed in a number of cultures. This can often be in the form of high volume consumption, with the associated risks of excess energy and alcohol intoxication. It is plausible that novel approaches to brewing could alter the view of beer to being like that of other fermented products in our culture, in a similar way to products like kefir and kombucha, which, although they have low alcohol content, are seen as health products with long lists of testimonials to their health benefits. With recent advances in brewing methods and the selection of yeasts, it is possible to make highly acceptable low alcohol beverages that retain the potential for health benefits. It may then be a language issue in that beer is associated in society with drunkenness, and perhaps fermented grain and hop drink may be more acceptable. It is also of interest that a number of crude spontaneous brewed beverages based on grains in Africa, e.g., "Ogi" are gathering increasing interest as potential therapies for diarrheal diseases. Therefore, akin to kefir and kombucha, low alcohol beers appear to have a significant place in traditional diets and medical practice. However, although there is considerable potential for an alcohol-free beer to have health benefits, it may not be acceptable to many traditional beer drinkers. This may be an issue of marketing. With an increasing awareness of health effects and a desire to have a non-soft drink when socializing but needing to avoid alcohol to drive has seen a growth in alcohol-free beer sales of 25% or more per year since 2017 according to GlobalData. It may be necessary to drop the beer name to reach a wider health market, but in view



of the potential growth of the market potentially co-existing with health potential, this appears to be a product area worthy of considerable investment and scientific, as well as clinical, investigation.

This increasing diversity to what is considered to be beer may not be entirely compatible with the views of some brewers and may be challenging at least to the principles of the German Purity Laws, as the ingredients will diverge from the four stated, and novel methods to minimize alcohol will require the further exploration of fermentation cultures. However, for industries including brewing to grow, new and sustainable approaches are needed to meet the needs of society, both with respect to the consumer and governmental policies and agendas. Therefore, innovation to produce a highly palatable product that can maximize health and minimize any potential harm seems to be a logical aspiration for a responsible brewing industry.

## 7. Conclusions

Despite the clear challenge that beer is often considered to be a source of empty calories and harmful alcohol, it also can contain a range of potentially health improving compounds. If beer can move beyond being a source of alcohol, beer could be a potentially interesting health product, being isotonic and containing a range of botanicals that have potential health benefits. This change is entirely possible with the growth of craft brewing processes that have explored the use of probiotic yeasts, including those that can produce a lower alcohol beer with a similar flavor and aroma profile whilst retaining the potential health benefits. As more of these new products start to appear, it is important that both scientific and clinical research is undertaken to assess the actual effects on human health. Consumers may be confused about alcohol-free beers in terms of whether they are an alternative to traditional beer or a health drink. This could require a change of name to lose the "beer" name; therefore, consumer research into both regular beer drinkers and health conscious consumers is required.

**Author Contributions:** Conceptualization, D.D.M. and R.C.; Writing—Original draft preparation, D.D.M.; Writing—Review and editing, R.C, B.H.-K.; Supervision, R.C. All authors have read and agreed to the published version of the manuscript.

**Funding:** This research received no external funding.

**Conflicts of Interest:** Raymond Carson is a Director of BeerTorrent Ltd. The other authors declare no conflict of interest.

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
