# Peer review of "A Review of the Potential Health Benefits of Low Alcohol and Alcohol-Free Beer: Effects of Ingredients and Craft Brewing Processes on Potentially Bioactive Metabolites"

_beverages, doi:10.3390/beverages6020025_

Round 1

Reviewer 1 Report

Please find below my comments:

some minor text editing is needed

the sentence ”was commonly consumed by both adults and children from at
least the 11th century in the UK” needs to be supported by a reference

The paragraph containing lines 38-42 - needs citing references

Lines 64-66 - will be useful for the reader to add some supporting numbers

Some important references related to the health component are missing: 

A comprehensive characterization of beer polyphenols by high resolution mass spectrometry (LC-ESI-LTQ-Orbitrap-MS). Quifer-Rada et al., 2015. Food Chemistry 

Hop-Derived Prenylflavonoids and Their Importance in Brewing Technology: A Review. Mudura, 2015, Bulletin UASVM Food Science and Technolog

Brown beer vinegar: A potentially functional product based on its phenolic profile and antioxidant activity. Mudura et al., 2018, Journal of the Serbian Chemical Society Wine, Beer, Alcohol and Polyphenols on Cardiovascular Disease and Cancer. Arranz et al., 2012. Nutrients. Characterization of bioactive compounds and antioxidant activity of fruit
beers, Nardini, 2020, Food Chemistry Risk Factors Associated with Alcohol Consumption Among Romanian University Students- Preliminary Research. Salanta et al., 2018. Bulletin UASVM Food Science and Technology. Beer and its non-alcoholic compounds in health and disease. Osorio-Paz et al., 2019. Critical Reviews in Food Science and Nutrition A New Perspective on the Health Benefits of
Moderate Beer Consumption: Involvement of the Gut Microbiota. Quesada-Molina et al., 2019. Metabolites. Simultaneous identification of low-molecular weight phenolic and nitrogen
compounds in craft beers by HPLC-ESI-MS/MS. Cheiran et al., 2019. Food Chemistry. A table containing bioactive compound, source, amount, health effect, daily dose needed, the average dose taken from the beer, and the citing reference will represent structured information for the reader.  Please also give some data related to the worldwide craft beer market. Who are the top consumer countries? Are there some health-related aspects - positive/negative? What is the limit for beer to be consumed as a daily recommended dose?  Please give evidence also about other chemical compounds that might have a toxic effect on human health? It is more useful to add clear numbers than describing paragraphs (e.g. ”Providing a source of probiotic microbes, which requires a viable dose of live organisms to be in the product when it is consumed.” - specify the dose), which in many points need the addition of supporting references. The manuscript does not offer enough evidence for the health benefits of craft beer consumption.  Too general Conclusion section.        

Author Response

We would like to thank the reviewer for their constructive and thorough comments, we have responded to each point individually.

some minor text editing is needed

We have worked through the manuscript and made a number of edits.

the sentence ”was commonly consumed by both adults and children from at
least the 11th century in the UK” needs to be supported by a reference

A reference has been added, sorry for this omission

The paragraph containing lines 38-42 - needs citing references

A reference has been added, sorry for this omission

Lines 64-66 - will be useful for the reader to add some supporting numbers

Data has been added

Some important references related to the health component are missing: 

A comprehensive characterization of beer polyphenols by high resolution mass spectrometry (LC-ESI-LTQ-Orbitrap-MS). Quifer-Rada et al., 2015. Food Chemistry 

Hop-Derived Prenylflavonoids and Their Importance in Brewing Technology: A Review. Mudura, 2015, Bulletin UASVM Food Science and Technolog

Brown beer vinegar: A potentially functional product based on its phenolic profile and antioxidant activity. Mudura et al., 2018, Journal of the Serbian Chemical Society Wine, Beer, Alcohol and Polyphenols on Cardiovascular Disease and Cancer. Arranz et al., 2012. Nutrients. Characterization of bioactive compounds and antioxidant activity of fruit
beers, Nardini, 2020, Food Chemistry Risk Factors Associated with Alcohol Consumption Among Romanian University Students- Preliminary Research. Salanta et al., 2018. Bulletin UASVM Food Science and Technology. Beer and its non-alcoholic compounds in health and disease. Osorio-Paz et al., 2019. Critical Reviews in Food Science and Nutrition A New Perspective on the Health Benefits of
Moderate Beer Consumption: Involvement of the Gut Microbiota. Quesada-Molina et al., 2019. Metabolites. Simultaneous identification of low-molecular weight phenolic and nitrogen
compounds in craft beers by HPLC-ESI-MS/MS. Cheiran et al., 2019. Food Chemistry.

Many thanks for your suggestions, these have been added to the manuscript along with a search for clinical trials involving beer from the WHO clinical trials database

A table containing bioactive compound, source, amount, health effect, daily dose needed, the average dose taken from the beer, and the citing reference will represent structured information for the reader. 

A table has been added, as this focuses on nutrients, with the addition of polyphenols which have had their health effects discussed in detail in the text the information with respect to the health effects were not included in the table. The daily dose needed was not included as this would vary by age, gender and is variable by country so this data would not be extremely meaningful. Instead a compromise to include data from common beverages which are sources of these nutrients were added.

Please also give some data related to the worldwide craft beer market. Who are the top consumer countries?

Data on the market size have been added

Are there some health-related aspects - positive/negative? What is the limit for beer to be consumed as a daily recommended dose? 

Details of alcohol recommendations have been added

Please give evidence also about other chemical compounds that might have a toxic effect on human health?

Although there are potential harms from some compounds including polyphenols, this was not within the scope of this review as unlike some products e.g. green tea extracts the concentrations found in beer are not high enough to be of concern.

It is more useful to add clear numbers than describing paragraphs (e.g. ”Providing a source of probiotic microbes, which requires a viable dose of live organisms to be in the product when it is consumed.” - specify the dose), which in many points need the addition of supporting references.

The potential of probiotics has been clarified

The manuscript does not offer enough evidence for the health benefits of craft beer consumption. 

This has been rephrased along with the conclusion to consider the terminology used with respect to beer and to look at low alcohol and alcohol free beers.

Too general Conclusion section.        

Clarification has been added

Reviewer 2 Report

A review of the potential health benefits of beer is very interesting work. However in reviewer opinion, it is doubtful to recognize the beer as a source of bioactive, pro-health compounds, due to its alcohol content. On the other hand, non-alcoholic beer (even very rich in bioactive compounds) will probably not be considered as an alternative to "drinking" society.

The authors very often used the phrase "moderate consumption". What does it mean? once a week? Or maybe twice a month? Probably it means something different for everyone, thus where is a limit, behind which we have the problem of alcoholism.

In reviewer opinion some comparison of the health benefit (polyphenols, vitamins, probiotics content, etc.) of beer and other beverages (yoghurt, etc.) is necessary. Comparison of beer composition (in form of table) and such type of competitive products, enable assessment of the actual benefits of beer consumption. For this reason, a lot of literature should be added in this paper regarding beer composition, e.g.:

Marova, I.; Parilova, K.; Friedl, Z.; Obruca, S.; Duronova, K. Analysis of Phenolic Compounds in Lager Beers of Different Origin: A Contribution to Potential Determination of the Authenticity of Czech Beer. Chromatographia 2011, 73, S83–S95. Socha R., PajÄ…k P., Fortuna T., Buksa K.: Antioxidant activity and the most abundant phenolics in commercial dark beers. International Journal of Food Properties, 20, 2017, 595-609. Zhao, H.; Chen, W.; Lu, J.; Zhao, M. Phenolic Profiles and Antioxidant Activities of Commercial Beers. Food Chemistry 2010, 119, 1150–1158. Pai, T.V.; Sawant, S.Y.; Ghatak, A.A.; Chaturvedi, P.A.; Gupte, A.M.; Desai, N.S. Characterization of Indian Beers: Chemical Composition and Antioxidant Potential. Journal of Food Science and Technology 2015, 52, 1414–1423.

Author Response

Thank you for your constructive comments we have attempted to respond to each of your points individually:

A review of the potential health benefits of beer is very interesting work. However in reviewer opinion, it is doubtful to recognize the beer as a source of bioactive, pro-health compounds, due to its alcohol content. On the other hand, non-alcoholic beer (even very rich in bioactive compounds) will probably not be considered as an alternative to "drinking" society.

Thank you for this point, we have attempted to respond to this through the text, and included data on the growth in the size of the market and how some of these products are being marketed and used.

The authors very often used the phrase "moderate consumption". What does it mean? once a week? Or maybe twice a month? Probably it means something different for everyone, thus where is a limit, behind which we have the problem of alcoholism.

This has been clarified using the UK guidelines.

In reviewer opinion some comparison of the health benefit (polyphenols, vitamins, probiotics content, etc.) of beer and other beverages (yoghurt, etc.) is necessary. Comparison of beer composition (in form of table) and such type of competitive products, enable assessment of the actual benefits of beer consumption. For this reason, a lot of literature should be added in this paper regarding beer composition, e.g.:

Marova, I.; Parilova, K.; Friedl, Z.; Obruca, S.; Duronova, K. Analysis of Phenolic Compounds in Lager Beers of Different Origin: A Contribution to Potential Determination of the Authenticity of Czech Beer. Chromatographia 2011, 73, S83–S95. Socha R., PajÄ…k P., Fortuna T., Buksa K.: Antioxidant activity and the most abundant phenolics in commercial dark beers. International Journal of Food Properties, 20, 2017, 595-609. Zhao, H.; Chen, W.; Lu, J.; Zhao, M. Phenolic Profiles and Antioxidant Activities of Commercial Beers. Food Chemistry 2010, 119, 1150–1158. Pai, T.V.; Sawant, S.Y.; Ghatak, A.A.; Chaturvedi, P.A.; Gupte, A.M.; Desai, N.S. Characterization of Indian Beers: Chemical Composition and Antioxidant Potential. Journal of Food Science and Technology 2015, 52, 1414–1423.

Thank you these, along with other references have been added to the manuscript

Reviewer 3 Report

This manuscript describes the potential health benefits of beer. The authors specifically address the hypothesis that developing a novel fermentation process based on beer technology it will be possible to obtain a fermented drink, with low or no-alcohol, containing probiotics and phytochemicals beneficial to human health. The main concern is that the Authors are referring to a fermented drink that does not exist yet and which is not clear if it could/should be labeled as “beer” without misleading consumers. Probiotics and phytoestrogens can be introduced into the diet using more conventional ways. Cell viability is a major challenge for the efficacy of probiotics drinks, and it is difficult to imagine a stable beer with high numbers of viable probiotics. As an example, the Japanese Beverages Association demands a minimum number of 107 colony-forming units (CFU) per milliliter of probiotic microorganisms at the end of shelf-life. One additional point that warrants special attention from the Authors is the final fermentation product(s) that should be produced either than ethyl alcohol: glycerol or fusel alcohols, lactic acid or other organic acids? As discussed in Reference 3 by Capece et al. there are other considerations that make the question more complex.
I strongly suggest the Authors focus their attention on the challenges to develop a low-car, low- or zero-alcohol fermented drink based on barley rather than on the health benefit of beer.

Minor comments
- Some sentences (ie line 42-48) should be reformulated to enhance readability.
- use a white space before the parenthesis-qua-character delimiting reference number citation in the text.
- Line 130, “it has been proposed”: the Authors should include a reference.
- Line 133-134, the Authors should include a reference.
- Line 138, modify in “inhibiting virus replication and bacteria and fungi growth” (viruses do not grow!).
- Line 211-212, the Authors should include a reference.
- Line 240-244, microbial species should be written in italics.
- Line 306 and 308, modify “Dekkra” in “Dekkera
- Line 442, pages are missing.
- Line 458, the reference is incomplete.

Author Response

Thank you for your constructive and thorough review, we have attempted to respond to each point individually:

This manuscript describes the potential health benefits of beer. The authors specifically address the hypothesis that developing a novel fermentation process based on beer technology it will be possible to obtain a fermented drink, with low or no-alcohol, containing probiotics and phytochemicals beneficial to human health. The main concern is that the Authors are referring to a fermented drink that does not exist yet and which is not clear if it could/should be labeled as “beer” without misleading consumers. Probiotics and phytoestrogens can be introduced into the diet using more conventional ways.

The types of alcohol free-beer which may fit this category including those already on the market have been included in the manuscript. This along with how these products might be presented to the consumer to avoid confusion.

Cell viability is a major challenge for the efficacy of probiotics drinks, and it is difficult to imagine a stable beer with high numbers of viable probiotics. As an example, the Japanese Beverages Association demands a minimum number of 107 colony-forming units (CFU) per milliliter of probiotic microorganisms at the end of shelf-life. One additional point that warrants special attention from the Authors is the final fermentation product(s) that should be produced either than ethyl alcohol: glycerol or fusel alcohols, lactic acid or other organic acids? As discussed in Reference 3 by Capece et al. there are other considerations that make the question more complex.
I strongly suggest the Authors focus their attention on the challenges to develop a low-car, low- or zero-alcohol fermented drink based on barley rather than on the health benefit of beer.

Thank you for this comment, we have added comments regarding the number of colony forming units. The difficulty with respect to maturing of products and secondary fermentation was briefly mentioned. However, with the suggestions from other reviewers to include more information with respect to polyphenols in beer and potential health effects, it would not be possible to include information on the complexity of metabolites of final fermentation and maintain a coherent manuscript. The point about the challenges of alcohol free beers, has been partially considered, but again with the comments of other reviewers focusing on health aspects it was not possible to manage this within the constraints of this manuscript.

Minor comments
Some sentences (ie line 42-48) should be reformulated to enhance readability.

This section has been edited

use a white space before the parenthesis-qua-character delimiting reference number citation in the text.

This has been altered, thank you

Line 130, “it has been proposed”: the Authors should include a reference.

Thank you this has been added 

Line 133-134, the Authors should include a reference.

Thank you a reference has been added

Line 138, modify in “inhibiting virus replication and bacteria and fungi growth” (viruses do not grow!).

Sorry, this has been changed

Line 211-212, the Authors should include a reference.

Sorry, this has been added

Line 240-244, microbial species should be written in italics.

These have been italicized 

Line 306 and 308, modify “Dekkra” in “Dekkera”

Thank you these have been changed

 Line 442, pages are missing.

These have been added

Line 458, the reference is incomplete.

Thank you, this has been checked

Round 2

Reviewer 2 Report

-

Author Response

Many thanks for your kind review of our revised manuscript

Reviewer 3 Report

References

In the text, most of the references are reported with superscript numbers; "reference numbers should be placed in square brackets [ ], and placed before the punctuation.

In the References section, numbers should be not placed in brackets and issue numbers should be omitted.

Taxonomy

Through the text (see page 5-7), leave a single space in the scientific names of species: i.e. "S.cerevisiae" should be changed in "S. cerevisiae".

Author Response

Many thanks for your kind and supportive review. We have considered your suggestions for revision and made the following changes linked to each of your points.

References

In the text, most of the references are reported with superscript numbers; "reference numbers should be placed in square brackets [ ], and placed before the punctuation.

Apologies we applied the ACS template to our reference manager, but this did not work, we have gone through the manuscript and manually adjusted this. 

In the References section, numbers should be not placed in brackets and issue numbers should be omitted.

Thank you, we have removed brackets and issue numbers from references

Taxonomy

Through the text (see page 5-7), leave a single space in the scientific names of species: i.e. "S.cerevisiae" should be changed in "S. cerevisiae".

Thank you, this has been adjusted throughout the manuscript.

Thank again for your time and thoughtful comments in reviewing our manuscript.